# Triage admission protocol with a centralized quarantine unit for patients after acute care surgery during the COVID-19 pandemic: A tertiary center experience in Taiwan

Chih-Ho Hsu[1,2☯], Chen-Lun Chiu[1☯], Yi-Ting Lin[1], Ann-Yu Yu[1], Yen-Te Kang[3], Michael Cherng[3], Yi-Hui Chen[1], Ting-Hui Chuang[4], Hsin-Yi Huang[4], Jwo-Luen Pao[4], Kuo-Hsin Chen[3,5]*, Chih-Hung Chang[4,6]*

1 Division of General Surgery, Department of Surgery, Far-Eastern Memorial Hospital, New Taipei City, Taiwan, 2 Institute of Hospital and Health Care Administration, National Yang Ming Chiao Tung University, Taipei, Taiwan, 3 Department of Surgery, Far-Eastern Memorial Hospital, New Taipei City, Taiwan, 4 Department of Orthopedics, Far-Eastern Memorial Hospital, New Taipei City, Taiwan, 5 Division of Electrical Engineering, Yuan Ze University, Taoyuan City, Taiwan, 6 Graduate School of Biotechnology and Bioengineering, Yuan Ze University, Taoyuan City, Taiwan

☯ These authors contributed equally to this work.
* chen.kuohsin@gmail.com (KHC); orthocch@mail.femh.org.tw (CHC)

**Data Availability Statement:** The data underlying the results presented in the study are provided in the supplemental information.

## Abstract

### Background

During the COVID-19 surge in Taiwan, the Far East Memorial Hospital established a system including a centralized quarantine unit and triage admission protocol to facilitate acute care surgical inpatient services, prevent nosocomial COVID-19 infection and maintain the efficiency and quality of health care service during the pandemics.

### Materials and methods

This retrospective cohort study included patients undergoing acute care surgery. The triage admission protocol was based on rapid antigen tests, Liat® PCR and RT-PCT tests. Type of surgical procedure, patient characteristics, and efficacy indices of the centralized quarantine unit and emergency department (ED) were collected and analyzed before (Phase I: May 11 to July 2, 2021) and after (Phase II: July 3 to July 31, 2021) the system started.

### Results

A total of 287 patients (105 in Phase I and 182 in Phase II) were enrolled. Nosocomial COVID-19 infection occur in 27 patients in phase I but zero in phase II. More patients received traumatological, orthopedic, and neurologic surgeries in phase II than in phase I. The patients' surgical risk classification, median total hospital stay, intensive care unit (ICU) stay, intraoperative blood loss, operation time, and the number of patients requiring postoperative ICU care were similar in both groups. The duration of ED stay and waiting time for acute care surgery were longer in Phase II (397 vs. 532 minutes, p < 0.0001). The duration

**Funding:** The author(s) received no specific funding for this work.

**Competing interests:** The authors have declared that no competing interests exist.

of ED stay was positively correlated with the number of surgical patients visiting the ED (median = 66 patients, Spearman's ρ = 0.207) and the occupancy ratio in the centralized quarantine unit on that day (median = 90.63%, Spearman's ρ = 0.191).

## Conclusions

The triage admission protocol provided resilient quarantine needs and sustainable acute care surgical services during the COVID-19 pandemic. The efficiency was related to the number of medical staff dedicated to the centralized quarantine unit and number of surgical patients visited in ED.

## Introduction

In late December 2019, the first cases of a novel coronavirus-induced pneumonia were reported in Wuhan, China [1]. The World Health Organization (WHO) officially designated this infectious disease as coronavirus disease 2019 (COVID-19) on February 12, 2020 [2]. Due to the lack of effective medical management in the early period of the outbreak, the COVID-19 pandemic has had an enormous impact worldwide, causing more than 200 million confirmed cases globally and more than 4 million deaths as at August 11, 2021, according to data from the WHO [3, 4].

Taiwan, a country neighboring China, did not develop large-scale local outbreaks at the beginning of the pandemic [5]. Non-pharmaceutical interventions (NPIs), such as widespread lockdowns, quarantines, social distancing, personal hygiene, and the use of facemasks were the main COVID-19 control measures [6]. However, after community transmission of COVID-19 was first reported on May 11, 2021, the disease soon spread into northern Taiwan [7].

During the COVID-19 surge, patients were reluctant to visit hospitals because of the potential for contracting infections in a hospital environment and the government's "stay at home" policy. However, the acute care surgery demand persisted and constituted most of our medical service during this time. Disease severity and unintended consequences are collateral damage caused by the COVID-19 pandemic. For instance, some studies have investigated the negative influence of the pandemic on the treatment of acute appendicitis, including delays in the time to consultation, longer durations of admission, and higher rates of readmission, complicated appendicitis, and severe peritonitis [8, 9]. Cano-Valderrama et al. also noticed that minor complications in acute care surgery were more common during the pandemic because of significantly delayed arrival at the emergency department and more severe patients undergoing surgery [10].

After a nosocomial COVID-19 outbreak in the Far East Memorial Hospital (FEMH), we reduced ambulatory surgeries, postponed elective surgeries, and deferred hospitalization if possible (Fig 1). Thus, the number of patients undergoing elective and ambulatory operations decreased dramatically in comparison with the previous year; however, the number of patients undergoing acute care surgery remained as stable as before. To address the persistent demand for acute care surgery during the pandemic, we designed a triage admission protocol with a centralized quarantine unit to manage patients after acute care surgery. We hypothesized that this system would significantly prevent another outbreak of nosocomial infections arising from patients and their caregivers after acute care surgery. Furthermore, we hoped that this system might also maintain the efficiency and quality of our health care service during the pandemics and decrease the duration of stay in the emergency department.

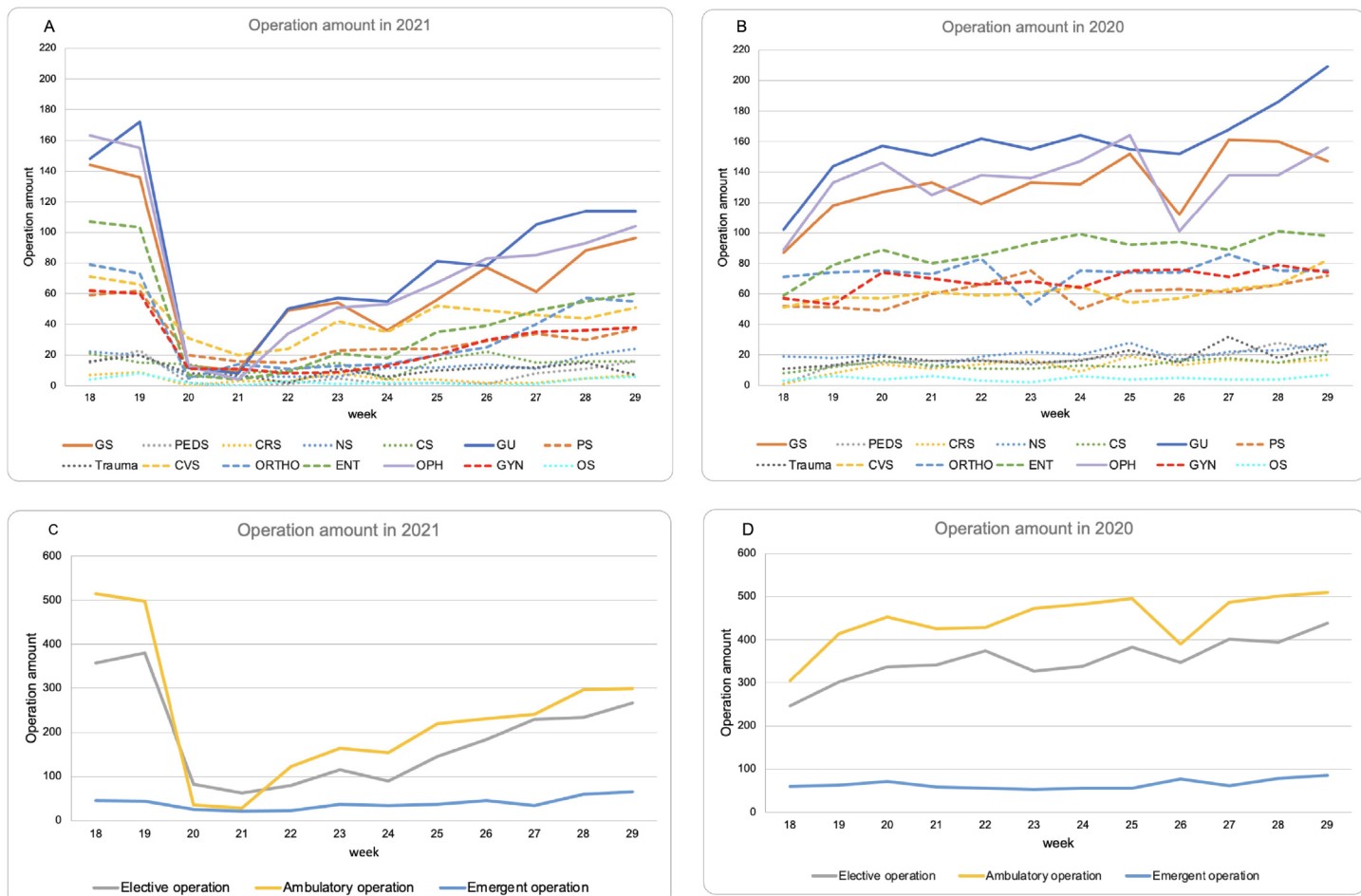

**Fig 1. Differences in the number of operations performed in the same period (between the 18th and 29th weeks) in 2020 and 2021.** A nosocomial infection occurred in FEMH on May 14, 2021, and policy of reducing surgical services was announced on May 16, 2021 (19th week). Panels A & B, number of operations stratified by department. Panels C & D, number of operations stratified by the nature of surgery (elective, ambulatory, or emergent surgery). GS, general surgery; PEDS, pediatric surgery; CRS, colorectal surgery; NS, neurosurgery; CS, chest surgery; GU, urological procedure; PS, plastic surgery; CVS, cardiovascular surgery; Ortho, orthopedic surgery; OPH, ophthalmological surgery; GYN, gynecological surgery; OS, oral and maxillofacial surgery.

## Materials and methods

### Study hospital characteristics

New Taipei City is situated in the northern part of Taiwan, and it has the largest population in Taiwan, with approximately 4 million inhabitants. FEMH is the only tertiary medical center in New Taipei City. FEMH is equipped with 1,383 beds and is visited by 6,500 outpatients each day, with approximately 400 patients treated each day in the emergency room, which is ranked fourth nationwide.

The first nosocomial infection was reported in FEMH on May 14, 2021. The nosocomial outbreak resulted in 27 confirmed cases, including 12 patients, 12 caregivers, and three nurses. Two ordinary ward units (total 88 beds) were isolated thereafter. Multidiscipline experts were appointed to the Infection Control and Special Response Committee. The committee addressed nosocomial outbreaks with methods such as increasing the intensive care unit (ICU) capacity, transforming the general ward facilities into dedicated COVID-19 wards, reorganizing physicians and nurses to new tasks, decreasing outpatient services, discouraging ward

visits, and shifting to online video or telephone calls. Meanwhile, our hospital was retained as a medical center for severe illness during the surge, treating 11% of the critical COVID-19 patients in Taiwan. As a result, we faced immense pressure in maintaining acute care surgical services during the pandemic, while protecting the medical staff and preventing new nosocomial infections.

## Detail of triage admission protocol

As described by Wake et al. [16], the Infection Control and Special Response Committee classified the wards in the FEMH into four zones and set up the entrance criteria of these zones: general wards, green zone; COVID assessment ward, yellow zone; COVID alert, red zone; and COVID, purple zone (see Fig 2). The four-tier triage system was used to stratify surgical patients and their caregivers to proper wards, regardless of whether the patients came from the outpatient department for elective operations or from the emergency department for acute care surgery. The innovation of our protocol is that suspicious or confirmed COVID-19 patients could also be stepped down to less guarded wards after completing isolation. To realize this idea, the Committee transformed a dedicated COVID care unit into a centralized quarantine unit that included yellow and red zones. The Committee also assigned a special medical team in the centralized quarantine unit to demonstrate management of patients after acute care surgery under this triage system. The team included three nurse practitioners and four residents to execute the protocol, and six attending physicians in different surgical subspecialities enrolled as the daily team leader.

RT-PCR testing had been indicated as the standard protocol in the diagnosis of COVID-19 [11], but analysis of specimens needed turnaround times of about 3.5 hours in our hospital during the pandemic. To ensure the efficiency and reliability of the triage admission protocol,

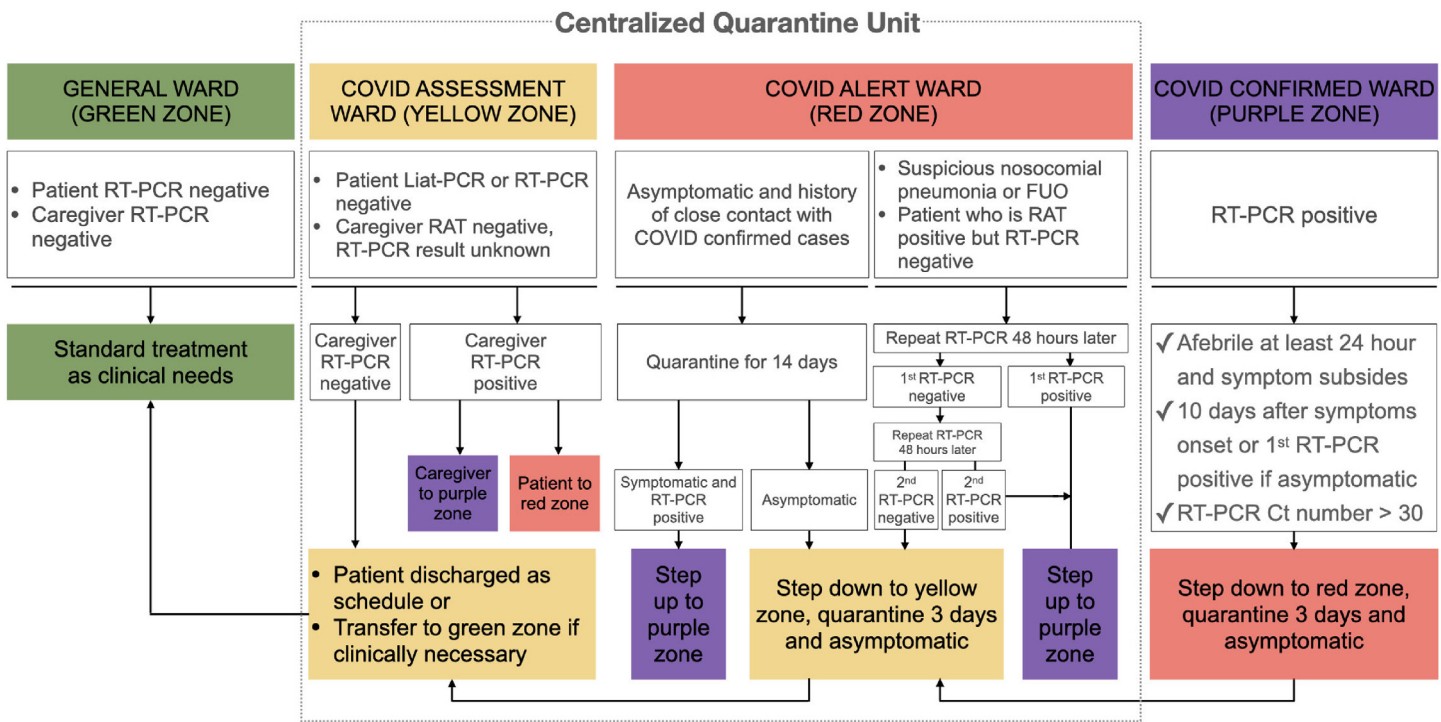

**Fig 2. Flowchart of the triage admission protocol with a centralized quarantine unit.** RAT, rapid antigen test; FUO, fever of unknown origin.

we incorporated timely and accurate laboratory tests into the system. The tests included the VTrust COVID-19 rapid antigen test (target antigen: SARS-CoV-2 nucleocapsid protein; reaction time: 15 min; positive predictive agreement: 93.1%; negative predictive agreement: 99.6% [12]) and the Cobas® Liat® PCR System (target genome region: RF1a/b and N genes; turnaround time: 20 min; positive predictive agreement: 100%; negative predictive agreement, 97.4% [13]). The Cobas® Liat® PCR System was only indicated for patients from the emergency department who needed acute care surgery. The National Health Insurance Administration of Taiwan funded these tests for inpatients and their caregivers during the pandemic.

Outpatients who were ready to undergo elective operations were first admitted to wards in the green zone. Patients and their caregivers provided nasopharyngeal swab samples for RT-PCR testing 2 days before the scheduled admission date and were admitted only if SARS-CoV-2 was not detected by RT-PCR. The patients subsequently received surgical services depending on their clinical needs. The Committee monitored occult COVID-19 infection in the hospital by performing rapid antigen tests on patients and their caregivers every 7 days.

Patients who underwent non-deferrable surgery due to acute disease (such as laparoscopic appendectomy for ruptured appendicitis, orthopedic fixation surgery for displaced fractures, herniorrhaphy for incarcerated hernia) were admitted to wards in the yellow zone. The patients underwent Liat® PCR testing at the outdoor COVID triage center in ED once consultants decided to arrange for a non-deferrable operation, and they were allowed to undergo surgery in an ordinary operation room if the Liat® PCR results were negative. The patient's caregiver was allowed entry into the hospital only after showing negative results in a rapid antigen test and was required to complete an RT-PCR test within 24 hours after entering the yellow zone. When patients and their family caregivers tested negative for SARS-CoV-2, the primary care medical team continued their treatment in the yellow zone ward or transferred them to the green zone ward if their admission was estimated to last longer than a week. We also discouraged changes in caregivers for patients who were in the green or yellow zones. If there was a change, the new family caregiver was required to undergo a self-paid RT-PCR examination (approximately 190 USD) to prove SARS-CoV-2 negativity.

The Committee prepared six COVID alert wards (red zone) for surgical patients who showed negative RT-PCR or Liat® PCR test results and met any of the following criteria:

- Patients who developed nosocomial pneumonia and were recommended quarantine by the Infection Control Committee

- Patients who developed fever of unknown origin (FUO) and were recommended quarantine by the Infection Control Committee

- Patients requiring perioperative care who showed positive results in the rapid antigen test and negative results in the RT-PCR test

- Patients who had been in close contact with COVID-confirmed people within the last two weeks

- Patients who were exposed to a cluster of COVID-19 infections among their co-residents

- Patients who needed perioperative care during their home isolation period

- Patients exposed to any specific COVID-19 outbreak area in the last two weeks

Patients admitted to the red zone due to contact history were quarantined in the red zone for 14 days to monitor symptoms of COVID-19 infection. RT-PCR testing was performed if the patient became symptomatic. For patients admitted to the red zone because of suspected pneumonia, FUO, or discrepancy between the results of the rapid antigen test and RT-PCR

test, the Committee recommended follow-up evaluations with at least one nasopharyngeal swab RT-PCR test at intervals of 48 hours. According to Lauer et al. [14], the incubation period of SARS-CoV-2 is estimated to be 3 to 7 days, with a range of 2 to 14 days, so the Committee decided to remove patients from quarantine gradually. Thus, patients in the red zone who completed the quarantine and remained RT-PCR-negative were stepped down to the yellow zone, monitored for 3 days, and then discharged or transferred to the green zone according to their clinical needs.

All RT-PCR-positive patients were placed in the COVID-confirmed ward (purple zone) where the patients were under the care of dedicated nurses and internists who specialized in infectious diseases and chest medicine. These patients were released from isolation when they met all of the following criteria announced by Taiwan CDC: (1) the patient is afebrile for at least 24 hours and symptoms subside or show improvement on chest radiographs, (2) at least 10 days have passed after the initial symptom onset or first positive RT-PCR result if the patient was initially asymptomatic, (3) one set of specimens from the respiratory tract (sputum, oropharyngeal, or nasopharyngeal swabs) showed negative results for SARS-CoV-2 or a Ct value > 30 in RT-PCR. Using the same principle of gradually removing patients from quarantine, patients from the purple zone were stepped down to the red zone, monitored for 3 days, further stepped down to the yellow zone, monitored for 3 days, and then discharged or transferred to the green zone according to clinical needs.

Critical surgical patients were transferred to ordinary ICU (green zone) or dedicated ICU (purple zone), depending on whether they were diagnosed with COVID-19 or not. We encouraged surgeons to communicate with intensivists by phone, and the intensivists in ICU were primarily responsible for managing patients' conditions.

To facilitate execution of the centralized quarantine unit and triage system, the special medical team also educated staff, patients, and caregivers on the ideal approaches to prepare themselves in the hospital. For example, the Committee strongly recommended that all medical staff wear basic personal protective equipment (PPE) in the green and yellow zones. If working in the red and purple zones or performing any aerosol-generating procedures, medical staff were required to upgrade to advanced PPE (see Fig 3). A special medical team in the centralized quarantine unit helped complete RT-PCR tests for family caregivers within 24 hours and treat patients in COVID alert wards. The team communicated with the surgeons whose patients were admitted in the yellow zone about the timing of discharge or transfer to the green zone. They also monitored febrile surgical patients in the green zone to identify those who might need to be transferred to the red zone if they were highly suspicious of COVID-19 infection. When patients were scheduled to undergo emergent operations, medical staff used shared decision-making tools to allow them to understand the benefits and risks of admission during the COVID-19 pandemics. We also provided posters and videos online (https://www.youtube.com/watch?v=EGkBic_r2ek) to educate patients and caregivers to maintain appropriate levels of hygiene during admission. All of these efforts were made to facilitate application of this triage system and decrease nosocomial COVID-19 transmission.

## Data collection

From May 11 to July 31, 2021, patients who underwent acute care surgery after evaluation in the emergency department (ED) were included in this study. We defined phase I as the period before the centralized quarantine unit was established (from May 11 to July 2), and phase II as the period after the centralized quarantine unit was established (from July 3 to July 31). A wide variety of acute care surgeries were included, as follows: general surgery, neurosurgery, plastic surgery, thoracic surgery, genitourinary surgery, colorectal surgery, orthopedic surgery,

| GENERAL WARD (GREEN ZONE) | COVID ASSESSMENT WARD (YELLOW ZONE) | COVID ALERT WARD (RED ZONE) | COVID CONFIRMED WARD (PURPLE ZONE) |
|---|---|---|---|

**Basic Personal Protective Equipment**

- Surgical mask
- N95 mask (optional)
- Face shield/Goggles

If **aerosol-generating procedures\*** need to be performed, use advanced personal protective equipment

**Advanced Personal Protective Equipment**

- N95 or higher respirator
- Surgical mask
- Face shield/Goggles
- Disposable surgical cap
- Gowns and coveralls
- Shoe covers
- Two-layer gloves

\* Aerosol-generating procedures are listed but not limited to: intubation, extubation, tracheotomy/tracheostomy procedures, cardiopulmonary resuscitation, induction of sputum, bronchoscopy, use non-invasive ventilation [Bi-level positive airway pressure ventilation (BiPAP), Continuous positive airway pressure ventilation (CPAP), high-flow nasal oxygen cannula]

**Fig 3. Recommendation of personal protective equipment for medical staff in different admission areas.**

traumatological surgery, pediatric surgery, otorhinolaryngologic surgery, ophthalmologic surgery, and gynecologic surgery. Patient demographic data, such as age, sex, operation codes, operation time, blood loss, demand for postoperative ICU care, and postoperative complications were documented. The American Society Anesthesiologists (ASA) physical status classification system was used as a simple categorization of a patient's status to help predict and record operative risk [15]. We also evaluated the indices related to administration efficacy, including time spent in the ED, number of surgical patients visiting the ED per day, number of patients admitted to and discharged from centralized quarantine units per day, length of general ward stay, length of ICU stay, length of total hospital stay, and occupancy and discharge ratios in the centralized quarantine units. The ED stay time was calculated from the time when the patient was registered in the ED to the time when the patient was transferred to the operating room or ward.

### Ethical statement

The study was ethically approved by the Institutional Review Board of FEMH (Reference FEMH No.: 110181-E.) Permission was sought from the hospital administration before data collection and analysis. Patient records and information was anonymized prior to analysis to ensure confidentiality of individual patient information. The ethics committee waived the requirement for individual patient consent due to the retrospective nature of the study.

### Statistics

Data were summarized as the median, interquartile range (IQR), and total range for continuous variables and as proportions for categorical variables. The Kolmogorov–Smirnov

goodness-of-fit test and normality plot were used to measure the distributional characteristics of the study variables. Since the target variables were not normally distributed, non-parametric statistical analysis was used to verify the proposed relationships. Thus, the Mann-Whitney U (Wilcoxon Rank Sum) test was used to compare the significant outcomes between the two independent groups. Spearman's rank correlation analysis was used to determine the correlation between continuous variables. All analyses were performed using SPSS Statistics for Windows (Version 22.0, IBM Corp, Armonk, NY, USA), and differences were considered statistically significant at $p < 0.05$.

## Results

A total of 287 patients received acute care surgery after evaluation at the ED of FEMH between May 11 and July 31, 2021 (Table 1). One hundred and five patients (36.6%) were admitted to the green zone in phase I, and 182 patients (63.4%) were admitted to the yellow zone in phase II. Twenty-seven cases of nosocomial COVID-19 developed in phase I but zero from patients or caregivers in the centralized quarantine unit, and three patients were transferred to the red zone (one from the green zone, and two from the yellow zone) due to the rapid onset of fever of unknown origin. All red zone patients were confirmed to be SARS-CoV-2 PCR-negative at least three times. Among the 182 patients who were admitted in yellow zone, 150 patients (82.4%) were discharged to home, 18 patients (9.9%) were transferred to green zone for prolonged medical needs, 12 patients (6.6%) were transferred to ICU postoperatively and two patients (1.1%) died due to severe surgical complications.

Among the 287 patients, orthopedic surgery was the most frequent acute care surgery during this period (n = 105, 36.7%), followed by general surgery (n = 104, 36.2%). The proportions of patients receiving traumatological, orthopedic, and neurologic surgeries increased significantly (500%, 418%, and 167%) from phase I to phase II. Most patients were classified as ASA II (n = 192, 66.9%). The unit operated efficiently, which was evidenced by the large number of patients admitted to and discharged from it (median, 16.5; IQR = 11,7), the high occupancy ratio (median, 90.63%; IQR = 12.5%), and the high discharge ratio (28.13%, IQR = 18.75%) (Table 2).

A comparison of the findings obtained in phase I and phase II showed no differences in the operation time (p = 0.1), blood loss (p = 0.426), length of total hospital stay (p = 0.736), and length of general ward stay (p = 0.619). Six patients needed intensive care after the operation in phase I and 15 patients required it in phase II. The length of ICU stay did not differ between the two phases (p = 0.461).

In phase II, the time of patient stay in the ED was significantly longer (397 vs. 532 min, $p < 0.001$) than that in phase I (Table 2). The number of surgical patients visiting the ED per day in phase II was also significantly higher (40 vs. 66, $p < 0.001$) than that in phase I. The longer staying time in the ED was positively correlated with the number of surgical patients visiting the ED (Spearman's ρ coefficient = 0.207) and the occupancy ratio in the centralized quarantine unit on that day (Spearman's ρ coefficient = 0.191).

## Discussion

The triage admission protocol and centralized quarantine unit ensured that the hospital was protected from a new outbreak of nosocomial COVID infections. It also allowed the hospital to increase the volume of inpatient services for patients undergoing non-deferrable traumatological, orthopedic, and neurologic surgeries. The quality of surgical service was considered to be stable because the operation time, intraoperative blood loss, length of hospital stay, and ICU stay were not statistically different between the two periods. Although the duration of ED

**Table 1. Basic characteristics of the patients.**

| Factors | Phase I | Phase II | Total | P-value* |
|---|---|---|---|---|
| | (n = 105) | (n = 182) | (n = 287) | |
| Age (years) | | | | |
| Median (range) | 51 (17–87) | 56 (17–89) | 54 (17–89) | 0.149 |
| Sex | | | | 0.003 |
| Male | 72 | 93 | 165 (57.5%) | |
| Female | 33 | 89 | 122 (36.2%) | |
| Surgery departments | | | | 0.518 |
| General surgery | 53 | 51 | 104 (36.2%) | |
| Neurosurgery | 3 | 8 | 11 (3.8%) | |
| Plastic surgery | 5 | 3 | 8 (2.8%) | |
| Thoracic surgery | 6 | 3 | 9 (3.1%) | |
| Genitourinary surgery | 9 | 10 | 19 (6.6%) | |
| Colorectal surgery | 2 | 0 | 2 (0.7%) | |
| Orthopedic surgery | 17 | 88 | 105 (36.7%) | |
| Traumatological surgery | 1 | 6 | 7 (2.4%) | |
| Pediatric surgery | 0 | 1 | 1 (0.3%) | |
| Otorhinolaryngologic surgery | 4 | 3 | 7 (2.4%) | |
| Ophthalmologic surgery | 3 | 3 | 6 (2.1%) | |
| Gynecologic surgery | 2 | 6 | 8 (2.8%) | |
| ASA classification [15] | | | | 0.095 |
| ASA I | 6 | 12 | 18 | |
| ASA II | 66 | 126 | 192 | |
| ASA III | 28 | 36 | 64 | |
| ASA IV | 2 | 1 | 3 | |
| Operation time (mins) | | | | |
| Median (Q1-Q3) | 60 (50–100) | 85 (60–120) | 70 | 0.1 |
| Estimated blood loss (ml) | | | | |
| Median (Q1-Q3) | 10 (5–50) | 10 (5–50) | 10 | 0.426 |
| Postoperative ICU care | 6 | 15 | 21 | 0.424 |
| Length of ICU stay (days) | | | | |
| Median (Q1-Q3) | 3 (3–8) | 3 (2–4) | 3 | 0.461 |
| Length of total hospital stay (days) | | | | |
| Median (Q1-Q3) | 4 (3–8) | 4 (3–7) | 4 | 0.736 |
| Length of general ward stay (days) | | | | |
| Median (Q1-Q3) | 4 (3–8) | 4 (3–6) | 4 | 0.619 |

ASA classification, The American Society Anesthesiologists (ASA) physical status classification system; ICU, intensive care unit.

* Results of the Mann-Whitney U test.

stay, which serves as an index of administrative efficiency, could not be reduced by this system, this factor was more related to the increasing demand for ED visits and inpatient services.

The reported experiences in hospitals in the UK and South Korea have highlighted the spread of nosocomial COVID-19 infections among medical workers and patients [16–18]. Some useful strategies that were applied in FEMH to address this problem are listed as follows: (1) regular SARS-CoV-2 PCR testing for medical workers in high-risk facilities, (2) isolation of diagnosed COVID-19 cases until they showed two negative samples (including nasopharyngeal swab, throat swab, or deep respiratory sputum on the same day) with complete resolution

**Table 2. Efficacy index of ED and the centralized quarantine unit.**

| Factors | Phase I | Phase II | Total | P-value[*] | Spearman's ρ |
|---|---|---|---|---|---|
| | (n = 105) | (n = 182) | (n = 287) | | |
| Length of ED stay (mins) | 397 (285–570) | 532 (340–1153) | 465 (318–872.5) | <0.0001 | 1 |
| Number of surgical patients visiting in ED per day | 40 (32–51) | 66 (61–74) | 59 (47–72) | <0.0001 | 0.207[§] |
| Number of patients utilizing centralized quarantine unit per day | N/A | 16.5 (9–20.7) | | | |
| Occupying ratio in centralized quarantine unit per day (%) | N/A | 90.63 (84.38–96.88) | | | 0.191[§] |
| Discharging ratio in centralized quarantine unit per day (%) | N/A | 28.13 (15.63–34.38) | | | -0.003 |

ED, emergency department.

[*]Result of Mann-Whitney U test.

[§]P-value < 0.05 in Spearman's rank correlation analysis.

of symptoms, (3) social distancing (1.5-meter rule in non-clinical areas) in staff areas and virtual meetings in the hospital; (4) use of proper PPE by medical staff in clinical practice; (5) administration of approved COVID-19 vaccines to all hospital staff and associated cooperation partners; and (6) availability of highly efficient and accurate SARS-CoV-2 PCR testing throughout the day. Caregivers, however, are another potential source of viral shedding because they move between community and healthcare facilities in pandemics. In Brazil, Passarelli et al. surveyed 150 asymptomatic visitors using RT-PCR with nasopharyngeal specimens in a single day, and six of the 150 (4%) asymptomatic visitors were diagnosed with COVID-19 at a hospital with a universal masking policy [19]. Two inpatients (contacts) subsequently developed symptoms.

In Taiwan, several healthcare facilities have reported nosocomial infections since May 2021, including a total of 137 medical staff, 104 patients, and 49 caregivers (Table 3). Caregivers form the medium for viral shedding between hospitals and communities. These experiences and evidence confirm that caregiver management is a key factor in preventing hospital transmission in the future.

Some COVID-19 admission triage systems have been reported and applied in the context of different clinical needs and hospital capacities. Wake et al. implemented a

**Table 3. Number of COVID-19 nosocomial infections (n = 290).**

| Type of personnel | |
|---|---|
| Patients | 104 |
| Caregivers | 49 |
| Health care workers | |
| Health care assistants | 83 |
| Nurses | 43 |
| Physicians | 11 |
| Type of health care institute | |
| Medical center | 63 |
| Regional hospital | 218 |
| District hospital | 3 |
| Clinic | 6 |

Summary of nosocomial COVID-19 infections in Taiwan between January 2020 and August 2021. Data are adapted from a webinar by Da-Cheng Qu, Taipei Municipal Hospital, 2021. https://www.youtube.com/watch?v=tOgEPFoUnjY. Permission for citation was obtained from the author.

clinical assessment tool based on rapid SARS-CoV-2 PCR testing, clinical history and findings, laboratory results, and radiologic results to improve the effectiveness of COVID-19 triage and admission. They categorized patients into four tiers—COVID 0 (negative/not suspected), COVID 1 (unlikely/low risk), COVID 2 (likely/high risk), and COVID 3 (positive)—and allocated these patients to the non-COVID ward, assessment ward, and COVID ward. Patients in the non-COVID ward were assessed for their daily COVID risk level, and they were upgraded to COVID 1/2/3 if the risk level increased [16]. Deora et al. proposed a flowchart to manage neurosurgical patients [20]. In our study, patients from the outpatient service had to undergo RT-PCR before admission for elective surgery, and they were admitted to the green zone ward perioperatively only if their RT-PCR test was negative. However, if the patients presented to the ED, they would receive a screening questionnaire, chest computed tomography, and rapid antigen test first as triage. Emergency surgery was indicated if the patient had non-deferable disease, and the rapid antigen test results classified patients into the red zone (rapid antigen test positive) or orange zone (rapid antigen test negative). The patients would subsequently undergo operation and postoperative care in dedicated units in the red and orange zones, respectively. On the other hand, patients who required elective or semi-emergent surgery would complete an RT-PCR test first and were admitted to the green zone if the RT-PCR test showed negative results. Donà et al. reported their pathways in the ED of a pediatric hospital to stratify children and their caregivers according to clinical characteristics, concomitant comorbidities, and epidemiological risk of COVID-19 [21]. To prevent in-hospital virus spread from asymptomatic children or caregivers, the authors suggested nasopharyngeal swab testing for both children and caregivers if hospitalization was needed. Our triage protocol used the strengths of these forementioned systems. The four-tier triage protocol was based on clinical condition, rapid COVID-19 antigen test, Liat® PCR System and RT-PCR testing. It could be applied in both elective and acute surgical patients according to clinical urgency. Inpatient's family caregivers were also evaluated in the protocol by rapid COVID-19 antigen tests and RT-PCR testing. The protocol also allowed patients to be upgraded or downgraded in different tiers, so that the medical resources could be utilized accordingly. Therefore, we considered that this novel triage admission protocol made our hospital sustainable and resilient in the COVID-19 pandemic, but extra laboratory exams would also increase prolonged ED stay when surgical patients from ED gradually increased [22].

At the beginning of the COVID-19 pandemic, all healthcare workers were reorganized and reassigned to new hospital tasks. For example, frontline ward nurses were assigned to dedicated COVID-19 wards, COVID rapid screening stations, and vaccine administration stations in communities, and specialized quarantine facilities outside the hospital. When nurses had different work contents, some discouraging factors such as detailed policies for infection control, long-term PPE discomfort, and frustration were negatively associated with nurses' work engagement [23]. In addition, since we planned to increase inpatient services for patients after acute care surgeries, nurses could not leave and return to their original working environment because of the persistence of the COVID-19 pandemic. Therefore, appropriate workload planning for the limited number of nurses who were available for inpatient services was an important part of planning for our centralized quarantine unit. For instance, to address the demand arising from a gradual increase in the number of surgical patients in the ED, we expanded the yellow zone from 20 to 32 beds after recruiting nurses from 5 to 10 people per day. Despite controlling the admission days and transferring patients to the green zone if they needed longer inpatient treatment, the occupancy ratio in the yellow zone remained high. The large number of patients entering and leaving the unit substantially increased the doctors' and nurses' workloads.

This study had some limitations. First, this triage admission protocol with a centralized quarantine unit was applied to a single well-organized private tertiary medical center in Taiwan. Administrative policies, hospital culture, and healthcare insurance payers may differ across healthcare facilities and countries; thus, the efficacy of this system may not be reproducible in other facilities. Second, the COVID-19 pandemic in Taiwan was rapidly controlled by multiple approaches. For example, the Central Epidemic Command Center (CECC) in Taiwan restricted NPIs and instead encouraged wide screening in specific communities to identify asymptomatic COVID-19 patients. Some medical centers, including FEMH, treated moderate and severe COVID-19 patients inside hospitals and also set up special quarantine facilities outside the hospitals to manage most mild symptomatic COVID-19 patients, thereby preserving healthcare capacity for other severe diseases. Most importantly, the CECC encouraged vaccination in the population of Taiwan with the help of other countries, including Japan, the United States, Lithuania, Slovakia, the Czech Republic, and Poland. The peak of this pandemic was reached on May 28, with 597 confirmed cases, and only 17 confirmed cases were reported on July 31 [24]. Thus, the absence of nosocomial infection in this study might be related to the low prevalence of COVID-19 in Taiwan during this time.

## Conclusion

In summary, we utilized a rapid antigen test, Liat® PCR, and RT-PCR tests as the basis for designing a triage admission protocol to manage patients from EDs who needed acute care surgery. Zero nosocomial infection occurred when we applied this triage admission protocol. The centralized quarantine unit addressed the quarantine requirements for both patients and family caregivers. The efficiency of the unit is related to the number of medical staff dedicated and number of surgical patients visited in ED. Overall, this system provided resilient quarantine needs and sustainable acute care surgical services in our hospital during the COVID-19 pandemic.

## Supporting information

**S1 Data.**
(XLSX)

## Acknowledgments

We are grateful to all the staff members who assisted in this study.

## Author Contributions

**Conceptualization:** Kuo-Hsin Chen.

**Data curation:** Chen-Lun Chiu, Yi-Ting Lin, Ann-Yu Yu, Yen-Te Kang, Michael Cherng, Yi-Hui Chen, Ting-Hui Chuang.

**Formal analysis:** Chih-Ho Hsu.

**Funding acquisition:** Kuo-Hsin Chen, Chih-Hung Chang.

**Project administration:** Chih-Ho Hsu, Chen-Lun Chiu, Yi-Ting Lin, Yen-Te Kang, Michael Cherng, Yi-Hui Chen, Ting-Hui Chuang, Hsin-Yi Huang.

**Resources:** Hsin-Yi Huang, Jwo-Luen Pao, Kuo-Hsin Chen, Chih-Hung Chang.

**Software:** Chen-Lun Chiu.

**Supervision:** Hsin-Yi Huang, Jwo-Luen Pao, Kuo-Hsin Chen, Chih-Hung Chang.

**Visualization:** Chih-Ho Hsu, Ann-Yu Yu.

**Writing – original draft:** Chih-Ho Hsu, Chen-Lun Chiu.

**Writing – review & editing:** Chih-Ho Hsu.

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
