## [Decision Letter · Decision Letter 0]

3 Dec 2021

PONE-D-21-34174Triage admission protocol with a centralized quarantine unit for patients after acute care surgery during the COVID-19 pandemic: A tertiary center experience in TaiwanPLOS ONE

Dear Dr. Chen,

Thank you for submitting your manuscript to PLOS ONE. After careful consideration, we feel that it has merit but does not fully meet PLOS ONE’s publication criteria as it currently stands. Therefore, we invite you to submit a revised version of the manuscript that addresses the points raised during the review process. Please re-organize your manuscript to let the reviewers and the general readers easily read it.

We look forward to receiving your revised manuscript.

Kind regards,

Etsuro Ito

Academic Editor

PLOS ONE

Journal Requirements:

4. Your abstract cannot contain citations. Please only include citations in the body text of the manuscript, and ensure that they remain in ascending numerical order on first mention.

Reviewers' comments:

Reviewer's Responses to Questions

**Comments to the Author**

1. Is the manuscript technically sound, and do the data support the conclusions?

Reviewer #1: Yes

Reviewer #2: Partly

2. Has the statistical analysis been performed appropriately and rigorously? 

Reviewer #1: Yes

Reviewer #2: No

3. Have the authors made all data underlying the findings in their manuscript fully available?

Reviewer #1: Yes

Reviewer #2: No

4. Is the manuscript presented in an intelligible fashion and written in standard English?

Reviewer #1: Yes

Reviewer #2: No

5. Review Comments to the Author

Reviewer #1: This study addressedopne method of triage admission protocol with a centralized quarantine unit for patients after acute care surgery during the COVID-19 pandemic in one medical center. I have somes suggestions listed below.

1. Please correct the mistake of including English-edition highlights.

2. Taiwan demonstrated the effective control of COVID-19 pandemic. From the experiences in one Taiwan medical center, other countried may learn something from Taiwan. Although this study illustrated one policy of allocation of the patients in need of acute care surgery, the clinical benefit is lacking. In addition to the number of nosocomial infection, I wonder if this novel triage contributes to the less burden of medical professionals or reduction of medical costs.

Further, what is your SOP for the surgical patients with confirmed COVID-19?

Reviewer #2: This article described the comprehensive practice handling acute surgery and pre and post OP care during COVID-19 outbreak in the FEMH. The experience was very precious and deserved recording. However, there are many parts that need to be clarified, as followed in detailed:

Major concerns:

1. First of all, the manuscript was not well-organized. It made the reviewer uncomfortable very much to review it.

2. The major drawback is no COVID-19 confirmed cases admitted to the unit during the study period. Thus, the study goal, to prevent nosocomial COVID-19 infection was not possible to achieve, despite the authors described in details about the implementation of triage admission protocol and centralized quarantine.

3. It’s suggested to describe the scale, involvement and duration of nosocomial outbreak in FEMH. Did the outbreak continue during the study periods?

4. The mean waiting time in ED was longer in phase 1 than phase 2 (397 vs. 532 minutes, p < 0.0001). Although the authors explained that’s positively correlated with the number of surgical patients visiting, it’s suggested to explain more for the triage admission policy. Moreover, there was two different screening methods employed, i.e., RT-PCR, and rapid Ag test + Liat PCR test, respectively. Above 2 methods could have different turn around time. Furthermore, the mean numbers of staff members could be different in these 2 periods, too. The authors may try to put more variables in to analysis.

5. In the table 1, the authors described mean ICU stay was 3 days in both groups Please clarify the zone color of ICU in FEMH according to the triage admission policy. Did the medical teams continue the ICU care? What is the level of PPE ICU?

6. The authors did not analyze the needed staff numbers for implementing triage admission and centralized quarantine.

7. The authors mentioned that hired foreign caregivers management is a key factor in preventing hospital transmission. Above view point was not relevant to the study results.

Other suggestions:

1. In Background part of Abstract, line 27-31. Suggest briefly describe why need to establish the system? For example: no space for surgery during pandemic period? Or patients receiving surgery had higher risk of COVID19 infection than before?

2. In Material and Method part of Abstract, line 33-39. Suggest briefly describe the composition of this system.

3. In Result part of Abstract, line 46-47. “The duration of ED stay and waiting time for acute care surgery were longer in Phase II (397 vs. 532 minutes, p < 0.0001)”: Does it mean the duration of ED stay and waiting time were worsen after the system? Suggest show the advantage of this system in the Result part of Abstract.

4. In Conclusion part of Abstract, line 52-57. Suggest briefly describe the advantage of this system.

5. In Introduction part, line 72-79. “after community transmission of COVID-19…online video or telephone calls.”. Is this part so-called “Triage admission protocol with a centralized quarantine unit”? If YES, these sentences should be put in the Result part. If NO, suggest delete these sentences since they had no correlation with the title.

6. In Introduction part, line 80-94. Suggest revise these sentences to focus on the frequent delay of surgery during pandemic period and the drawback of delay of surgery.

7. In Introduction part, line 100-104, Is there any reference for this system?

8. In Method part, line 132-134, the wards were classified into four parts, is there any reference for this classification? And who decided how to classify these wards?

9. In Method part, line 139, about the COVID care unit, please describe the member of this unit.

10. In Method part, line 140, about the medical team, please describe the member of this team.

11. In Method part, line 147-148. What is the relationship between “reliability of the triage system” with “timely and accurate laboratory tests”?

12. In Method part, line 156-236 “Outpatients who were….. decrease nosocomial COVID-19 transmission..” The authors should describe which part is the regulation of the Taiwan government? Which part consult the reference? And which part is the novel innovation of the hospital?

13. In Table 1. Suggest to provide P-value for Age, Gender, Surgery department, and ASA classification.

14. There should be reference and brief description of ASA classification in Method part and Table 1.

15. In Result part, line 301-331. The authors should try to show the advantage of this “triage admission protocol”. Is there any part of Phase II better than Phase I ?

16. In Discussion part, line 335-338. “The triage admission protocol……and neurologic surgeries.” Are these conclusions proper according to the result of this study? For example, how to prove “The triage admission protocol and centralized quarantine unit secured the hospital”?

17. In Discussion part, line 363-366 “Caregivers form the…. hospitals.” Please cite the reference.

18. Table 3, Line 375, “Data are adapted from a webinar by Da-Cheng Qu, Taipei Municipal” Is there any copy-right issue?

19. Is Discussion part, line 378-405. Please compare the advantage and disadvantage between these references and the result of this study.

20. In Discussion part, line 406-444. Is there any correlation between these discussions with the result of this study? If No, may consider delete these sentences.

6. PLOS authors have the option to publish the peer review history of their article (what does this mean?). If published, this will include your full peer review and any attached files.

Reviewer #1: No

Reviewer #2: No

---

## [Author Response · Author response to Decision Letter 0]

1 Jan 2022

Dear reviewer:

We are appreciated for your recommendation for the manuscript. All of your concern are replied in the file "Response to reviewers," and the manuscript is revised according to your precious comment. 

Best regards, 

Dr. ChihHo Hsu and Dr. KuoHsin Chen

---

## [Decision Letter · Decision Letter 1]

17 Jan 2022

PONE-D-21-34174R1Triage admission protocol with a centralized quarantine unit for patients after acute care surgery during the COVID-19 pandemic: A tertiary center experience in TaiwanPLOS ONE

Dear Dr. Chen,

Thank you for submitting your manuscript to PLOS ONE. After careful consideration, we feel that it has merit but does not fully meet PLOS ONE’s publication criteria as it currently stands. Therefore, we invite you to submit a revised version of the manuscript that addresses the points raised during the review process.

We look forward to receiving your revised manuscript.

Kind regards,

Etsuro Ito

Academic Editor

PLOS ONE

Journal Requirements:

Reviewers' comments:

Reviewer's Responses to Questions

**Comments to the Author**

1. If the authors have adequately addressed your comments raised in a previous round of review and you feel that this manuscript is now acceptable for publication, you may indicate that here to bypass the “Comments to the Author” section, enter your conflict of interest statement in the “Confidential to Editor” section, and submit your "Accept" recommendation.

Reviewer #1: All comments have been addressed

Reviewer #2: All comments have been addressed

2. Is the manuscript technically sound, and do the data support the conclusions?

Reviewer #1: Yes

Reviewer #2: Partly

3. Has the statistical analysis been performed appropriately and rigorously? 

Reviewer #1: Yes

Reviewer #2: N/A

4. Have the authors made all data underlying the findings in their manuscript fully available?

Reviewer #1: Yes

Reviewer #2: Yes

5. Is the manuscript presented in an intelligible fashion and written in standard English?

Reviewer #1: Yes

Reviewer #2: Yes

6. Review Comments to the Author

Reviewer #1: (No Response)

Reviewer #2: Thanks for the dedicated reply from the authors of the article ” Triage admission protocol with a centralized quarantine unit for patients after acute care surgery during the COVID-19 pandemic: A tertiary center experience in Taiwan”

Still, there are some suggestions the authors need to further address and revise in this manuscript, as followed:

1. The authors described the disadvantages of the longer duration of ED stay and waiting time for acute care surgery in Phase II, and correlated the phenomenon to the number of surgical patients visiting the ED and the occupancy ratio in the centralized quarantine unit in the Results. However, no further discussion about the true cause of long ED stay was addressed in the Discussions. Finally in the Conclusions, the authors stated the efficiency was related to the number of medical staff dedicated to the centralized quarantine unit. Please revise and make a logical statement according to the points we emphasize above.

2. Please provide more data regarding the occupancy of purple zone in centralized quarantine unit during phase II period.

3. In the abstract, it is suggested that:

(1) Add more in the background about ” Why need the centralized quarantine unit and triage admission protocol?” (line 28-31)

(2) Provide the correct number of “nosocomial COVID-19 infection” in phase I and II. (line 42-50)

(3) Provide more statements about the infection control or clinical significance of the centralized quarantine unit and triage admission protocol, especially which outcome is better after intervention (phase II compared to phase I).

4. In Line 71-73 “However, after community transmission of……into northern Taiwan.” Please cite the reference.

5. In the Method part, it is suggested to highlight what is the novel innovation of the protocol in this study?

6. In Line 311-318 “Case 1……. and suspicious pneumonia.” Please clarify what is the significance of these three patients?

7. In the Result part, please state more clearly about which outcome are better in phase II than phase I.

7. PLOS authors have the option to publish the peer review history of their article (what does this mean?). If published, this will include your full peer review and any attached files.

Reviewer #1: No

Reviewer #2: No

---

## [Author Response · Author response to Decision Letter 1]

23 Jan 2022

Thanks for all comments from reviewers, and the comments were responded in the document ‘Response to Reviewers. ’

---

## [Editor Report · Decision Letter 2]

25 Jan 2022

Triage admission protocol with a centralized quarantine unit for patients after acute care surgery during the COVID-19 pandemic: A tertiary center experience in Taiwan

PONE-D-21-34174R2

Dear Dr. Chen,

We’re pleased to inform you that your manuscript has been judged scientifically suitable for publication and will be formally accepted for publication once it meets all outstanding technical requirements.

Kind regards,

Etsuro Ito

Academic Editor

PLOS ONE

---

## [Editor Report · Acceptance letter]

27 Jan 2022

PONE-D-21-34174R2 

Triage admission protocol with a centralized quarantine unit for patients after acute care surgery during the COVID-19 pandemic: A tertiary center experience in Taiwan 

Dear Dr. Chen:

I'm pleased to inform you that your manuscript has been deemed suitable for publication in PLOS ONE. Congratulations! Your manuscript is now with our production department. 

Kind regards, 

on behalf of

Prof. Etsuro Ito 

Academic Editor

PLOS ONE